# Degradation of the Tumor Suppressor PDCD4 Is Impaired by the Suppression of p62/SQSTM1 and Autophagy

**DOI:** 10.3390/cells9010218

**Published:** 2020-01-15

**Authors:** M. Manirujjaman, Iwata Ozaki, Yuzo Murata, Jing Guo, Jinghe Xia, Kenichi Nishioka, Rasheda Perveen, Hirokazu Takahashi, Keizo Anzai, Sachiko Matsuhashi

**Affiliations:** 1Department of Internal Medicine, Saga Medical School, Saga University, 5-1-1 Nabeshima, Saga 849-8501, Japan; monirbio31@gmail.com (M.M.); guojing198564@hotmail.com (J.G.); xiajinghe@hotmail.com (J.X.); rashu_bcmb@yahoo.com (R.P.); takahas2@cc.saga-u.ac.jp (H.T.); akeizo@cc.saga-u.ac.jp (K.A.); matsuha2@cc.saga-u.ac.jp (S.M.); 2Health Administration Centre, Saga Medical School, Saga University, 5-1-1 Nabeshima, Saga 849-8501, Japan; 3Division of Histology and Neuroanatomy, Department of Anatomy and Physiology, Saga Medical School, Saga University, 5-1-1 Nabeshima, Saga 849-8501, Japan; murata@iuhw.ac.jp; 4Division of Molecular Genetics and Epigenetics, Department of Biomolecular Sciences, Saga Medical School, Saga University, 5-1-1 Nabeshima, Saga 849-8501, Japan; pcg.trxg@gmail.com

**Keywords:** autophagy, p62/SQSTM1, PDCD4, proteasome, ubiquitination

## Abstract

PDCD4 (programmed cell death 4) is a tumor suppressor that plays a crucial role in multiple cellular functions, such as the control of protein synthesis and transcriptional control of some genes, the inhibition of cancer invasion and metastasis. The expression of this protein is controlled by synthesis, such as via transcription and translation, and degradation by the ubiquitin-proteasome system. The mitogens, known as tumor promotors, EGF (epidermal growth factor) and TPA (12-*O*-tetradecanoylphorbol-13-acetate) stimulate the degradation of PDCD4 protein. However, the whole picture of PDCD4 degradation mechanisms is still unclear, we therefore investigated the relationship between PDCD4 and autophagy. The proteasome inhibitor MG132 and the autophagy inhibitor bafilomycin A1 were found to upregulate the PDCD4 levels. PDCD4 protein levels increased synergistically in the presence of both inhibitors. Knockdown of p62/SQSTM1 (sequestosome-1), a polyubiquitin binding partner, also upregulated the PDCD4 levels. P62 and LC3 (microtubule-associated protein 1A/1B-light chain 3)-II were co-immunoprecipitated by an anti-PDCD4 antibody. Colocalization particles of PDCD4, p62 and the autophagosome marker LC3 were observed and the colocalization areas increased in the presence of autophagy and/or proteasome inhibitor(s) in Huh7 cells. In ATG (autophagy related) 5-deficient Huh7 cells in which autophagy was impaired, the PDCD4 levels were increased at the basal levels and upregulated in the presence of autophagy inhibitors. Based on the above findings, we concluded that after phosphorylation in the degron and ubiquitination, PDCD4 is degraded by both the proteasome and autophagy systems.

## 1. Introduction

PDCD4 (programmed cell death 4), also known as antineoplastic factor, contains two MA3 domains homologous to the M1 domain of eukaryotic translation initiation factor 4G (eIF4G), RNA binding domain, and nuclear localization signal (NLS) domains. PDCD4 binds to eIF4A and diminishes its helicase activity and thus inhibits the cap-dependent translation [1,2,3]. The RNA binding domain of PDCD4 binds to the internal ribosomal entry site (IRES) or secondary structure in the coding region of some mRNAs and inhibits the IRES-mediated translation or the elongation of protein synthesis [4,5]. PDCD4 is also considered a transcription inhibitor due to its ability to bind the p65 subunit of nuclear factor-kappa B (NF-κB) in human glioblastoma cells [6], Twist1 [7], and specific protein (Sp) family transcription factors [8]. PDCD4 suppresses the activation protein 1 (AP1) transcription activities, by inhibiting the activation of the upstream kinase mitogen activated protein kinase kinase kinase kinase 1 (MAP4K1) [3].

Previous studies have revealed the cell cycle inhibitory and apoptosis-inducing roles of PDCD4, so PDCD4 has been regarded as a tumor suppressor [9,10,11,12]. The downregulation of PDCD4 is observed in different types of cancer, such as colorectal carcinoma [13], prostate cancer [14], glioblastoma [15], lung cancer [16], and hepatocellular carcinoma [12]. Several mechanisms underlying the downregulation of PDCD4 have been reported. The PDCD4 mRNA expression is silenced by methylation at the 5′CpG island of the *PDCD4* promotor region [17]. However, *PDCD4* gene mutations have not been reported and the expression is mostly controlled post transcriptionally at translation and protein degradation levels [18]. MicroRNAs (miRNAs), which are endogenous non-coding small RNAs, bind to the 3′-UTR region of the transcript and induce either translational suppression or degradation of the mRNAs. Several miRNAs, including miR21, have been implicated in the regulation of the expression of PDCD4 and suppression of cancer cell apoptosis [19,20,21,22]. miR21 binds to the miR21 binding site localized at nt238-249 of the PDCD4 3′-UTR region and inhibits the translation [22,23]. EGF (epidermal growth factor) activates the PI3K (phosphoinositide 3-kinase)-AKT (protein kinase B)-mTOR (mechanistic target of rapamycin)-p70S6K1(ribosomal protein S6 kinase beta-1) signaling pathway. The activated p70S6K1 then phosphorylates PDCD4 and stimulates the degradation of the protein in the ubiquitin-proteasome system [24]. PDCD4 protein contains the SCF^βTRCP^ binding motif 71DSGRGD76S. As 71S and 76S in the degron are phosphorylated, PDCD4 protein is ubiquitinated by SCF^βTRCP^ ubiquitin ligase and degraded by the proteasome system. The phosphorylation of the upstream serine 67 (67S) triggers the phosphorylation of 71S and 76S [18,24]. TPA (12-*O*-tetradecanoylphorbol-13-acetate), a potent stimulator of protein kinase C (PKC), also induces the phosphorylation and degradation of PDCD4 protein. The TPA-activated PKCδ and PKCε signaling pathways phosphorylate PDCD4, leading to its degradation by the proteasome system in Huh7 hepatocellular carcinoma cells [18,25,26]. During the investigation of PDCD4 degradation mechanisms, we found that the proteasome inhibitor MG132 slightly upregulated the PDCD4 protein levels in Huh7 cells [26]. It led us to assume the presence of different regulatory pathways of this protein.

Macro-autophagy is a cellular process in which cytosolic protein and damaged organelles are sequestered by double-membrane vesicles called autophagosomes. ATG conjugation systems including ATG5 are required for the autophagosome formation. Mature autophagosomes fuse with lysosomes to have their contents degraded [27,28,29]. LC3-I, upon conjugation with phosphatidyl ethanol amine, makes LC3-II, which is considered an autophagy marker. During autophagosome biogenesis, LC3-II integrates with the autophagosome membrane [30]. p62 is an adaptor protein that binds to polyubiquitinated protein and brings them to bind with LC3-II in autophagosomes for degradation [31,32,33,34]. p62 carrying an ubiquitinated protein is subsequently degraded with substrate and LC3-II. This is considered as autophagic flux [30,35].

The EGF-activated nutrient responsive kinase mTOR is able to reduce autophagic flux [36,37,38]. ULK1 (unc-51-like kinase 1) is one of the key components of autophagosome biogenesis [39]. Under nutrient-rich conditions, the mTORC1 (mechanistic target of rapamycin complex 1) phosphorylates ULK1 and Atg13 and leads to the formation of ULK-Atg13-FIP 200 complex to reduce the kinase activity of ULK1 [38,40,41]. In contrast, under nutrient starvation condition or in the presence of mTOR inhibitor rapamycin, the autophagy system is initiated by stimulated ULK1 [42,43].

Autophagy is reportedly activated in many types of cancer and contributes to the survival of cancer cells, including HCC (hepatocellular carcinoma) cells [44,45,46]. The expression of the tumor suppressor PDCD4 is downregulated in many types of cancer. While the mechanisms underlying PDCD4 downregulation in cancer cells have been reported, whether or not the PDCD4 protein levels are regulated by autophagy is unclear. We therefore investigated the role of autophagy in the PDCD4 degradation pathway.

## 2. Materials and Methods

### 2.1. Cell Culture

The human hepatoma cell line Huh7 was obtained from the Japanese Cancer Research Resources Bank (Osaka, Japan). Cells were cultured and maintained in Dulbecco’s modified Eagle’s medium (DMEM) of Sigma-Aldrich (St. Louis, MO, USA) containing 10% fetal bovine serum, 100 µg/mL penicillin and streptomycin in 5% CO_2_ at 37 °C. A total of 3 × 10^5^ cells were seeded in 35-mm dishes and cultured for 72 h, and then the culture medium was replaced with fresh medium according to the experimental purposes.

### 2.2. Development of ATG5 Mutant-16 Huh7 Cell Line

CRISPR/Cas9 plasmid construction and cell transfection: to disrupt the *ATG*5 expression in Huh7 cells, *ATG*5-specific single-guide RNA (sgRNA) was designed using the online tool CRISPR DESIGN (http://CRISPR.mit.edu). The sgRNA targeting sequence was as follows: 5′-AACTTGTTTCACGCTATATC-3′ in exon 2 of the *ATG5* gene. Custom sgRNA targeting oligonucleotides were synthesized by Hokkaido System Science Co., Ltd. (Hokkaido, Japan). The CRISPR/Cas9 vector was the pRSI9 derivative (Cellecta, Inc., 320 Logue Ave, Mountain View, CA 94043 USA), in which the PCR-cloned Cas9 open reading frame and the sgRNA sequence backbone had been inserted (Addgene plasmids #41815 and #41824). The sequencing primer (pRSI_R1) was 5′-TACAGTCCGAAACCCCAAAC -3′.

According to the sgRNA targeting of *ATG*5 exon site, the *ATG*5 Crispr PCR identification primer was designed. The primer sequences were as follows: F 5′-CTTTGGTTGAAATAAGAATTTAGCCTG-3′, R 5′-AAGGTTAAATATCCCATTTGCCAC-3′. The PCR conditions were as follows: reaction at 94 °C for 2 min followed by 35 cycles of denaturation at 94 °C, 30 s; annealing at 55.6 °C, 30 s; extension at 72 °C, 1 min. The *ATG*5-mutant plasmid was transfected into Huh7 cells using Lipofectamine LTX of Life Technologies (Rockville, MD, USA) according to the manufacturer’s instructions. Subsequently, transfected Huh7 cells were treated with puromycin (20 μg/mL). Positive clones were confirmed by Western blotting to identify the *ATG5* knockout effects.

### 2.3. Reagents

The growth factor EGF was from R&D Systems (Minneapolis, MN, USA). TPA and bafilomycin A1 were purchased from Sigma-Aldrich. Rapamycin and MG132 were purchased from Calbiochem (San Diego, CA, USA). 3-metyladenine was the product of Adipo Gen Life Sciences (San Diego, CA, USA). Protein assay kits and Sure Beads Protein A Magnetic Beads were obtained from Bio-Rad Laboratories, Inc. (Hercules, CA, USA). Magnetic Racks were purchased from Invitrogen (Waltham, MA, USA). Protease Inhibitor Cocktail Tablets (Complete Mini) were purchased from Roche Diagnostic GmbH (Mannheim, Germany). RNAiso Plus was obtained from Takara (Kusatsu, Japan), High Capacity cDNA Reverse Transcription Kit and Power Up SYBR Green Master Mix were the products of Thermo Fisher Scientific (Waltham, MA, USA).

### 2.4. Antibodies

An anti-PDCD4 antibody was prepared by immunizing rabbits with a synthetic peptide corresponding to the N-terminal amino acid sequence [12]. This antibody was used for the Western blotting analyses. Antibodies against β-actin were purchased from Cell Signaling Technology (Beverly, MA, USA). Anti-PDCD4 (Human) polyclonal antibody (pAb) (PD024), guinea pig anti-p62 C-terminal pAb antibody (PM066), rabbit anti-p62 (SQSTM1) pAb antibody (PM045), rabbit anti-Atg5 pAb antibody (PM050), and mouse anti-LC3 monoclonal antibody (mAb) (M152-3) were obtained from MBL (Tokyo, Japan). Anti-ubiquitin antibody (ab7780) and donkey anti-mouse I_g_G H&L (DyLight650) antibody (ab96878) were purchased from Abcam (Cambridge, UK). Alexa Flour 488 donkey anti-rabbit I_g_G (H+L) antibody was obtained from Thermo Fisher (Waltham, MA, USA). Alexa Flour 555-conjugated donkey anti-guinea pig I_g_G (H+L) antibody (bs-0358D-A555) was obtained from Bioss Antibodies Inc. (Woburn, MA, USA). Anti PDCD4 mouse monoclonal antibody (sc-376430) was obtained from Santa Cruz Biotechnology, Inc., (Dallas, TX, USA). The antibodies were used according to the protocols provided by the respective companies.

### 2.5. Transfection of Plasmids

Huh7 cells were cultured for 4 days and then transfected with *PDCD4* and *GFP-PDCD4* plasmids [12] using Lipofectamine LTX of Invitrogen (Waltham, MA, USA) according to the manufacturer’s protocol.

### 2.6. Western Blotting Analyses

The collected cells were extracted by sonication in lysis buffer containing 50 mM Tris-HCl (pH 6.8), 2.3% sodium dodecyl sulphate (SDS) and 1 mM phenylmethylsulfonyl fluoride (PMSF). The cell debris was eliminated by centrifugation at 12,000× *g* for 10 min, and the supernatant was collected. Protein amounts were determined with a *DC*^TM^ protein assay kit of Bio-Rad Laboratories, Inc. (Hercules, CA, USA) using bovine serum albumin as the standard by the Lowry method. Protein (15-30 µg) from each sample was mixed with SDS loading buffer, separated by SDS polyacrylamide gel electrophoresis, and transferred to a polyvinylidene difluoride (PVDF) membrane (Bio-Rad). The membrane was blocked via incubation overnight at 4 °C in phosphate-buffered saline (PBS) containing 0.1% Tween 20 and 10% skim milk and then incubated with the primary antibody with shaking for 1 h at room temperature or overnight at 4 °C. After washing five times with PBS containing 0.1% Tween 20, the specific bands were visualized by further incubation with horseradish peroxidase (HRP)-conjugated second antibody followed by enhanced chemiluminescence detection using the ECL system (Amersham, Buckinghamshire, UK) according to the manufacturer’s instructions. The β-actin antibody was used as a control. The stained membrane was exposed to Fuji Medical X-ray film (Tokyo, Japan) and the specific protein bands were determined with the Image J software program (https://imagej.nih.gov/ij/) and normalized by β-actin.

### 2.7. siRNA-Mediated Knockdown of p62

A total of 2–3 × 10^5^ cells were cultured in 35-mm dishes and used for transfection experiment at 80–90% confluency levels. siRNA transfection was performed using Lipofectamine RNAiMAX (Life Technologies) according to the manufacturer’s protocols. The SQSTM1-2 (S100057596), SQSTM1-5 (S103089023), SQSTM1-6 (S103116750), SQSTM1-7(S103117513), and negative control (1027281) siRNAs were obtained from Qiagen (Heiden, Germany). The cells were collected for Western blotting analyses after 24 h of transfection.

### 2.8. Immunoprecipitation

After 4 h starvation in the presence of 10 µM bafilomycin A1 and 20 µM MG132, approximately 1 × 10^7^ cells/100-mm dish were lysed with 1 mL cold lysis buffer (50 mM Tris-HCl p^H^ 7.5, 150 mM NaCl, 0.05 % NP-40) containing 1 mM Complete Mini protease inhibitor. The cell suspension in the lysis buffer was incubated at 4 °C with rotation for 1 h, briefly sonicated (15–20 s) in the presence of ice, and then centrifuged at 12,000× *g* for 10 min at 4 °C. The supernatant was transferred to another fresh tube, and the protein concentration was determined by protein assay. Sure Beads Protein A Magnetic Beads and Magnetic Racks were used for the immunoprecipitation and isolation of specific protein targets. Immunoprecipitation of 500–700 µL lysate was performed using 3–5 μg anti-PDCD4 rabbit polyclonal antibody (PD024). Elution of the beads was carried out using SDS buffer (50 mM Tris-HCl pH 6.8, 2.3% SDS and 1 mM PMSF) with 10 min incubation at 70 °C. Finally, the purified target proteins were resolved by Western blotting analyses.

### 2.9. Quantitative Real-Time Reverse Transcription Polymerase Chain Reaction (qRT-PCR)

Total RNA from treated cells was isolated by using RNAiso Plus and reverse transcribed to cDNA using a High Capacity cDNA Reverse Transcription kit according to the manufacturer’s protocol. Quantitative Real-Time PCR (qRT-PCR) using Power Up SYBR Green Master Mix was performed on Step One Plus system of Applied Biosystems-Thermo Fisher Scientific (Waltham, MA, USA). The primers of GAPDH and PDCD4 were synthesized by Hokkaido System Science Co., LTD. (Hokkaido, Japan). The sequences of primers were as follows: GAPDH forward (F) 5′-GTCTCCTCTGACTTCAACAGCG-3′ and reverse (R) 5′-ACCACCCTGTTGCTGTAGCCAA-3′; PDCD4, (F) 5′-ATGAGCAGATACTGAATGTAAAC-3′ and (R) 5′-CTTTACTTCCTCAGTCCCAGCAT-3′. Data were analyzed using the comparative C_T_ (ΔΔC_T_) method, and the expression of target gene was normalized by GAPDH. Each experiment was repeated at least three times.

### 2.10. Microscopy and Imaging Analyses

For the colocalization analyses, 1.5 × 10^5^ cells were seeded onto glass coverslips (Matsunami Glass Co., Osaka, Japan) in 35-mm dishes and cultured in DMEM + 10% FBS medium. At 80–90% confluency, the cells were transfected with *PDCD4* plasmid and cultured for a further 24 h. For bafilomycin A1 or MG132 treatment, the cells were transfected with *PDCD4* plasmid and cultured for a further 20 h, and then the inhibitors were added at a final concentration of 10 µM bafilomycin A1 or 20 µM MG132. For control cells, the same amount of DMSO was used. For FBS absence, the dishes were washed twice with DMEM before adding the new medium. After 4 h of the addition of inhibitors, the cells were fixed in 4% paraformaldehyde by incubating 20 min at room temperature. Before fixation the cells were washed thrice with 1× PBS. The fixed cells were blocked at room temperature for 30 min with 1% bovine serum albumin and 1% donkey serum in phosphate-buffered saline. Incubation with primary antibodies was done overnight at 4 °C followed by 1 h incubation at room temperature with donkey anti-mouse IgG H&L (DyLight650) (ab96878) (Abcam), Alexa Flour 488 donkey anti-rabbit IgG (H+L) (Invitrogen), Alexa Flour 555 conjugated donkey anti-guinea pig IgG (H+L) (bs-0358D-A555) (Bioss Antibodies Inc.) secondary antibodies. One compartment from each treatment group was incubated with a secondary antibody and considered as blank. 4, 6-diamidino-2-phenylindole (DAPI) dihydrochloride (Dojindo, Kumamoto, Japan) was used for nuclear staining. Fluorescence images were captured using a confocal microscope (LSM880; Carl Zeiss, Oberkochen, Germany) at 20 and 63 (oil) magnifications. The images were processed and viewed using the Zen software program (Carl Zeiss, Oberkochen, Germany). All images were taken at 22 ± 3 °C. The captured images were analyzed using the HALO-2 image analyzing software program (Indica Labs, Albuquerque, NM, USA).

### 2.11. Statistical Analyses

Differences were determined using Student’s *t*-test, and *p* < 0.05 was considered significant. All of the experiments were performed at least in triplicate unless stated otherwise. Data are shown as the mean ± standard deviation (SD).

## 3. Results

### 3.1. The Autophagy Inhibitors Bafilomycin A1 and 3-Methyladenine Upregulated PDCD4 in Huh7 Hepatoma Cells

We previously showed that PDCD4 protein was phosphorylated at S67, S71, and S76 via the tumor promotor EGF and TPA-mediated signaling pathways and degraded by the ubiquitin-proteasome system [25,26]. While investigating the degradation mechanisms of PDCD4 protein mediated by tumor promotors, we repeatedly observed that EGF upregulated the PDCD4 protein levels in the presence of the proteasome inhibitor MG132 in Huh7 hepatoma cells, as shown in Figure 1. The result suggests the presence of PDCD4 degradation pathway(s) other than the proteasome-mediated pathway. Furthermore, EGF upregulated the autophagy-related factors ATG5 and p62 in an MG132 independent manner under both FBS-supplemented and FBS-deprived conditions (Figure 1). This phenomenon led us to hypothesize that PDCD4 protein may also be degraded by the autophagy system.

Huh7 cells were treated with the potent autophagy inhibitor bafilomycin A1 [47,48], a vacuolar ATPase (V-ATPase) inhibitor that prevents lysosomal acidification and substrate degradation [49,50]. As shown in Figure 2A and Appendix A, after treatment with bafilomycin A1, the levels of PDCD4 were increased significantly in time- and dose-dependent manners. The autophagy-related components p62, LC3-II, and ATG5 were also upregulated that was consistent with the previous report [51]. The PDCD4 protein levels similarly increased in both normal (+FBS) and autophagy-induced (-FBS) cultures (Figure 2A) in the presence of bafilomycin A1, indicating that the autophagy system was functioning properly in the normal culture without the induction of autophagy in Huh7 cells.

We also assessed the effects of 3-methyladenine (3-MA), another kind of autophagy inhibitor, on PDCD4 degradation in time-dependent manner. We found that the PDCD4 levels were upregulated in Huh7 cells treated with 3-MA compared to the control cells, but the autophagy-related proteins p62, ATG5, and LC3-II did not show significant accumulation in the cells (Figure 2B). Figure 2C shows that the proteasome inhibitor MG132 (M), the mTOR inhibitor rapamycin (R), and the autophagy inhibitor bafilomycin A1 (B) upregulated the PDCD4 levels compared to control cells. We also assessed the effects of these inhibitors in different combinations. As Figure 2C shows, the upregulatory effects of the combined treatments of B+R, B+M, and R+M were greater than that of any single inhibitor alone. MG132 as well as bafilomycin A1 upregulated the autophagy-related factors, but rapamycin did not increase the levels of p62 or LC3. This may have been due to enhanced autophagic flux, as rapamycin induces autophagy [52]. These results suggest that PDCD4 protein levels may be controlled by both proteasome and autophagy systems in Huh7 hepatoma cells.

### 3.2. The Autophagy Inhibitors Upregulated the PDCD4 Levels in Huh7 ATG5 Mutant Cells

To clarify whether or not PDCD4 protein was degraded by the autophagy system, an *ATG5* mutant cell line (*ATG5*-16) of Huh7 cells was isolated using the CRISPR-Cas9 system with screening for the ATG5-ATG12 protein band by Western blotting (Figure 3A). ATG5-ATG12 conjugates work like E-3 ligase to conjugate LC3-I to lipid phosphatidyl ethanolamine (PE) and thereby form LC3-II in the growing autophagosome membrane [53,54]. LC3-II formation was not observed by Western blotting in the *ATG5*-16 mutant cells (Figure 3A left panel). Furthermore, by immunocytochemistry LC3 particles, a marker of autophagosomes, were also not observed in the mutant cells upon starvation (Figure 3B, lower panel). The PDCD4 protein levels were higher in the mutant cells than in wild-type Huh7 cells, as shown in Figure 3A (right panel). The PDCD4 levels in both wild and *ATG5* mutant Huh7 cells were upregulated by TGFβ1 treatment, indicating that PDCD4 synthesis was not impaired.

We investigated the effects of the autophagy inhibitors bafilomycin A1 and 3-MA on PDCD4 degradation in *ATG5* mutant cells. Both inhibitors upregulated PDCD4 in the mutant cells as well as wild-type Huh7 cells (Figure 4A,B). Rapamycin and MG132 upregulated PDCD4 in a manner similar to that seen in wild-type cells, and EGF did not decrease the PDCD4 levels in the presence of MG132 in the mutant cells (Figure 4A). p62 and LC3-II proteins in Huh7 cells were accumulated in a time-dependent manner (Figure 2A), but the p62 levels in the *ATG5* mutant cells were higher than in the wild-type cells and unchanged in the presence of bafilomycin A1. On the other hand, the conversion of LC3-I to LC3-II was impaired in *ATG5* mutant cells. These results suggest that ATG5/ATG7-independent alternative macro-autophagy may be induced in ATG5-deficient Huh7 cells, as previously reported [55].

### 3.3. The PI3K Inhibitor 3-Methyladenine Upregulated the PDCD4 Level by Inhibiting Autophagy in Both Wild-Type and ATG5 Mutant Huh7 Cells

3-methyladenine (3-MA) prevents autophagosomes’ formation by inhibiting PI3K. The reagent is expected to inhibit the PDCD4 degradation in the ubiquitin-proteasome system because such degradation is induced by the mitogen-activated PI3K-Akt-mTOR-S6K1 signaling system [56,57]. Although TPA does not activate S6K1 in Huh7 cells, it induces phosphorylation of serines in PDCD4 degron by a different signaling pathway and leads to the degradation of this protein by ubiquitin proteasome system [26]. Therefore, we evaluated the effect of 3-MA on the PDCD4 levels in the presence of TPA. As shown in Figure 5, both MG132 and 3-MA upregulated PDCD4 although the upregulation by MG132 was not significant. In the absence of TPA, the upregulation by 3-MA was much higher than that induced by MG132 and was similar to that in the presence of both MG132 and 3-MA. TPA downregulated the PDCD4 levels in the presence of 3-MA, and the PDCD4 levels were restored in the presence of both 3-MA and MG132. These results suggest that TPA may induce PDCD4 degradation in both autophagy and proteasome systems. A similar result was obtained in ATG5 mutant cells although the upregulation by the inhibitor was not significant, which show a less marked degree of autophagy induction than wild-type Huh7 cells.

### 3.4. p62 is Involved in the Degradation of PDCD4

As mentioned earlier [31,32,33] p62/SQSTM1 binds with polyubiquitinated proteins and brings them to autophagosomes. We therefore performed siRNA-mediated p62 knockdown to assess the role of p62 in PDCD4 degradation. Initially we tested four different kinds of p62 (SQSTM1) siRNAs (SQSTM1-2, SQSTM1-5, SQSTM1-6, and SQSTM1-7) to find a suitable one and SQSTM1-2 and SQSTM1-6 were identified as the most effective (Figure 6A). Therefore, in the subsequent experiments, we used these siRNAs.

Figure 6 shows that the knockdown of p62 upregulated PDCD4 with both SQSTM1-2 and SQSTM1-6 siRNAs, indicating that p62 was involved in PDCD4 degradation. The PDCD4 protein levels in the p62 knockdown cells were slightly increased in the presence of the proteasome inhibitor MG132 but was not prominently increased in the presence of the lysozyme inhibitor bafilomycin A1, and the PDCD4 protein levels were increased even more in the presence of both reagents (Figure 6B). These results are consistent with the notion that PDCD4 protein may be degraded by both autophagy and proteasome systems after ubiquitination. In the *ATG5* mutant cells, p62 silencing also upregulated the PDCD4 protein levels, as shown in Figure 6C. This suggests that PDCD4 may be degraded by the autophagy system in the mutant cells as well as in the wild-type Huh7 cells.

For further investigation of the involvement of p62 in the degradation of PDCD4 protein, the extracts of wild-type and *ATG5*-16 mutant Huh7 cells were immunoprecipitated by anti-PDCD4 antibody, and the resulting precipitates were analyzed by Western blotting. The band of p62 as well as the smear bands of ubiquitin were found in the precipitates of both wild-type and mutant cells (Figure 6D). An LC3-II band but not an LC3-I band was found in the immunoprecipitation of wild-type Huh7 cells, while no band of either proteins was detected in the ATG5-deficient cells. These results indicate that the ubiquitinated PDCD4-p62 complex binds with LC3-II molecule located on the autophagosome vesicle (Figure 6D).

### 3.5. PDCD4 mRNA Levels Are Not Correlated with the Protein Levels

Previously it was shown that EGF slightly downregulated PDCD4 mRNA levels but TPA did not change the mRNA levels despite the mitogens downregulating the protein levels [25,26]. In this study we have shown that the autophagy inhibitor bafilomycin A1 and 3-MA upregulated PDCD4 levels. While bafilomycin A1 did not change mRNA levels, 3-MA upregulated the mRNA levels at 6 h treatment but downregulated at 9 h (Appendix A) although the PDCD4 levels were upregulated at the same treatment in the same cell line as shown in the Figure 2B. The results showed that the upregulation of PDCD4 protein levels by bafilomycin A1 was not concerned with the transcription levels but 3-MA upregulated PDCD4 at least partly by upregulating the mRNA levels. More detailed analyses are necessary to get conclusive results on the control mechanisms of PDCD4 expression at the transcription level.

### 3.6. PDCD4, p62, and LC3 Are Colocalized in Particles

Next, to determine whether or not p62-ubiquitinated PDCD4 complex transfers to autophagosomes, the colocalization of p62, PDCD4, and LC3 was investigated by immunocytochemistry using Huh7 cells. As shown in Figure 7, particles of the autophagosome marker LC3 were formed constitutively in the normal culture of Huh7 cells and increased in number and size when starved by serum removal and the colocalization of PDCD4, p62, and LC3 was observed under both normal and starved conditions (Figure 7A). The colocalization areas in the particles were slightly larger in the starved cells than in the normal culture cells (Figure 7B,C). These results indicated that autophagy was constitutively activated in Huh7 cells, which was consistent with the results of Western blotting showing that the autophagy inhibitors bafilomycin A1 and 3-MA upregulated PDCD4 levels under normal culture condition. The number and size of the colocalized particles were increased in the presence of bafilomycin A1 and/or MG132 (Appendix A).

In the *ATG5* mutant-16 cells, despite no LC3 particles being formed, the formation of p62 and PDCD4 colocalized particles were observed in the culture under normal conditions and the size of the particles was increased under starvation conditions (Figure 8). Furthermore, the number and size of the particles were increased in the presence of autophagy inhibitors bafilomycin A1 and/or proteasome inhibitor MG132 as observed in the wild-type Huh7 cells (Appendix A). The appearance of the particles was similar to that of wild type Huh7 cells (Appendix A). This may happen in an ATG5/ATG7-independent manner in ATG5 deficient Huh7 cells [55,58].

## 4. Discussion

In this article we demonstrated that PDCD4 protein is guided by p62 to bind with LC3-II on autophagosomes for degradation by the autophagy system. Selective autophagy is important as a backup system for substrates that cannot be degraded by the proteasome system, as they may be aggregated or the proteasome system is inactivated [59]. Thus, the cellular cytotoxic effect is alleviated by the synchronization of the proteasome and autophagy systems [60]. Mizushima et al. reported that the lysosomal protein degradation levels were suppressed in *Atg5* mutant cells in the presence of 3-MA [61]. Similarly, our present results showed that 3-MA increased the PDCD4 protein levels in *ATG5*-mutant Huh7 as well as in wild-type Huh7 cells (Figure 5). In wild-type Huh7 cells, 3-MA might inhibit PDCD4 degradation by blocking autophagosome formation.

Regarding how 3-MA upregulated the PDCD4 protein levels in the *ATG5* mutant cells, one possible mechanism is that 3-MA inhibits the PI3K-dependent phosphorylation and proteasomal degradation of PDCD4. We previously showed that TPA was able to induce the degradation of PDCD4 by the ubiquitin-proteasome system by activating the PKC signaling pathway independent of the PI3K pathways [25]. The findings from the present study showed that 3-MA and the proteasome inhibiter MG132 synergistically upregulated the PDCD4 levels in the presence of TPA in *ATG5*-mutant as well as wild-type Huh7 cells, suggesting that PDCD4 protein might be degraded by the macro-autophagy system, even in ATG5-deficinet Huh7 cells. As ATG5-deficient cells are unable to synthesize LC3-II from LC3-I, resulting in no formation of LC3 particles, macro-autophagy was expected to be inhibited in the mutant cells. However, both of the autophagy inhibitors bafilomycin A1 and 3-MA upregulated the PDCD4 levels in the mutant cells as well as wild-type Huh7 cells. Nishida et al. reported that Atg5/Atg7-independent alternative macro-autophagy is induced in *Atg5*^−/−^ MEF (Mouse Embryonic Fibroblasts) cells [55]. They also reported that the autophagosome membrane is derived from the vesicle of trans-Golgi and late endosome in *Atg5*^−/−^ or *Atg7*^−/−^ cells. However, Tsuboyama et al. conversely reported that autophagosome-like structures positive for the ER marker synthaxin-17 are formed in the absence of the ATG conjugation system [27]. At present, we cannot exclude the probability of involvement of different pathway(s) in addition to macroautophagy to degrade PDCD4 protein. So, further investigation is necessary to elucidate the entire mechanisms underlying PDCD4 protein degradation.

While it was initially suggested that autophagy prevents tumor initiation by maintaining cellular homoeostasis [46,62,63], autophagy has been reported to be upregulated in many types of cancer and contributed to the survival of cancer cells, including HCC [44,45,46,64]. However, the mechanisms by which autophagy supports tumor growth are not fully understood. Autophagy has been shown to be involved in the pro-tumorigenic processes of metabolic alterations, cell proliferation, and metastasis [65,66,67]. Saito et al. reported that p62-mediated autophagy was activated in HCC samples and promoted the proliferation of cancer cells and chemoresistance via the p62-mediated activation of Nrf2, which subsequently activates antioxidant genes [45].

Previous results showed that EGF and TPA were able to downregulate the PDCD4 levels. EGF and TPA activate the PI3k-Akt-mTOR-S6K1 axis and PKC, respectively, to phosphorylate PDCD4 at S71 and S76 in the degron and may channel it after ubiquitination to proteasomal degradation [24,25,26]. In this report, we found that the tumor suppressor PDCD4 was degraded by the p62-mediated autophagy system, showing that autophagy is at least one of the mechanism(s) underlying the PDCD4 downregulation in cancer cells. Our results also showed that the inhibition of autophagy increased the levels of PDCD4 in HCC. Recent clinical studies for cancer treatment have attempted to therapeutically target autophagy [68,69]. The increased levels of PDCD4 by autophagy inhibition might be involved in suppressing tumor development and progression and contribute to the treatment of various cancers. Other mechanism(s) besides proteasome and macro-autophagy systems might be associated with controlling the PDCD4 levels. However, further studies are necessary for understanding the entire regulatory mechanisms of this protein.

We demonstrated for the first time that the tumor suppressor PDCD4 is degraded by the p62-mediated selective macro-autophagy system in Huh7 hepatoma cells. The autophagy system may contribute at least partly to suppress the levels of PDCD4 and result in the development and progression of tumor cells. Thus, the inhibition of this pathway might be a potential target in cancer therapy.

## Figures and Tables

**Figure 1 cells-09-00218-f001:**
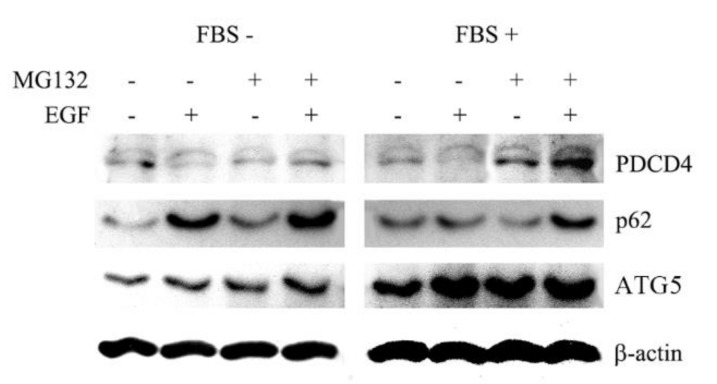
EGF (epidermal growth factor) upregulated the PDCD4 (programmed cell death 4) levels in the presence of MG132. Huh7 cells were cultured for 4 days and treated for 4h with or without 20 µg/mL EGF in the presence or absence of 20 µM MG132 under normal (+FBS) and serum-depleted (-FBS) culture conditions. The treated cells were then subjected to Western blotting analyses. Experiments were repeated at least three times and similar results were obtained from each experiment. A representative result is shown in the figure.

**Figure 2 cells-09-00218-f002:**
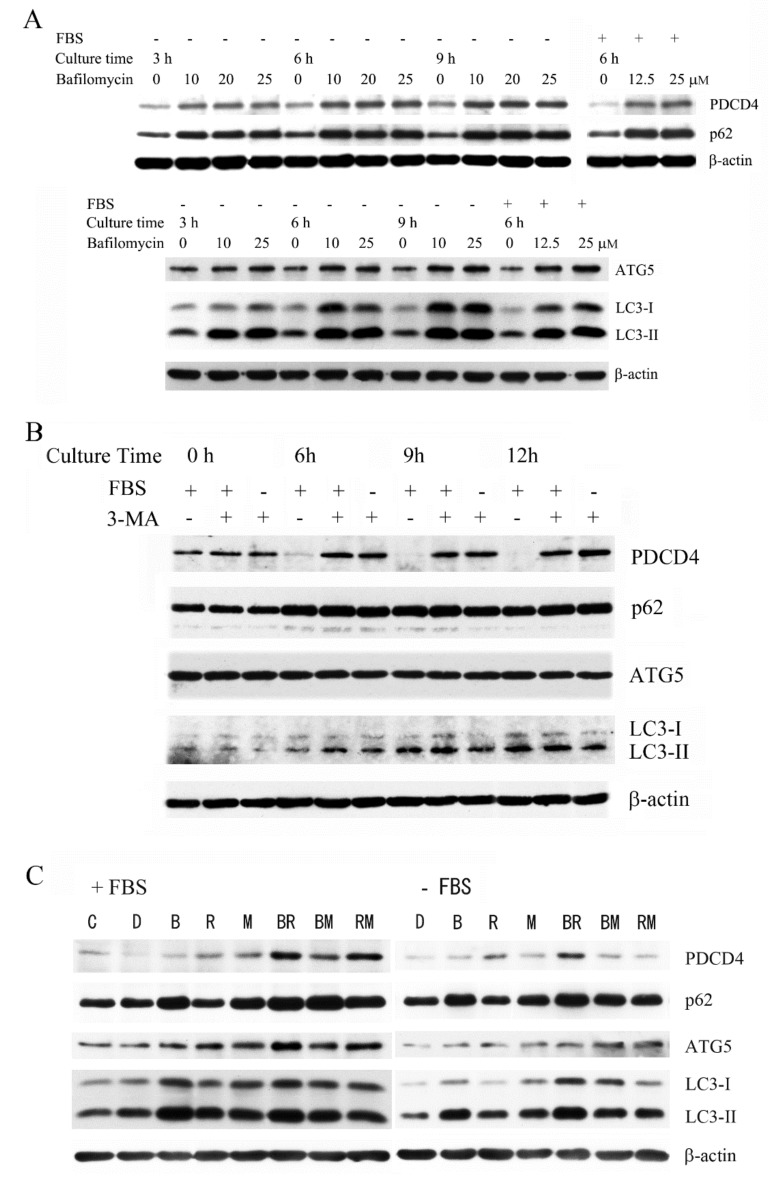
Both autophagy and proteasome inhibitors upregulated the PDCD4 levels. (**A**) After culturing for 4 days, Huh7 cells were treated with different concentrations of bafilomycin A1 as indicated in the figure, in the presence or absence of FBS for 3, 6, and 9 h. Western blotting was performed using antibodies against the components mentioned in the figure. The same sample was used to identify the components indicated in both the upper and lower panels. (**B**) Approximately 3 × 10^5^ Huh7 cells were cultured in a 35-mm dish in normal medium. At 90%–95% confluency, the medium was replaced with a medium either with or without FBS. The cells were treated with 5 mM 3-methyl adenine (3-MA) and collected after 6, 9, and 12 h for Western blotting. For analyses at 0 h, the cells were collected immediately after treatment (<15 min). (**C**) After culturing for 4 days Huh7 cells were treated with 10 µM bafilomycin A1 (B), 0.1 nM rapamycin (R), 20 µM MG132 (M), the combination of bafilomycin A1 and rapamycin (BR), bafilomycin A1 and MG132 (BM), rapamycin and MG132 (RM), and DMSO (D) as a control. After 4 h of treatment, the cells were collected for Western blotting. Experiments were repeated more than three times and similar results were obtained consistently in each case. The representative results are demonstrated.

**Figure 3 cells-09-00218-f003:**
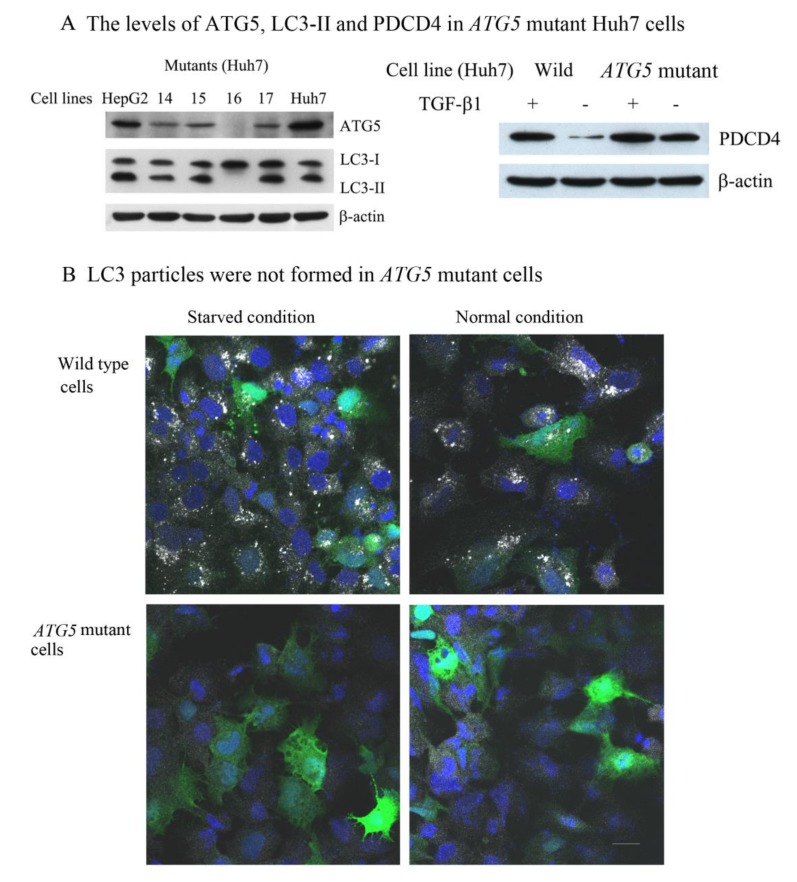
Characterization of *ATG5* mutant Huh7 cells (**A**) Left panel: the mutant 16 (*ATG5*-16) was unable to synthesize ATG5 and LC3-II molecules. Right panel: the PDCD4 levels were higher in the *ATG5* mutant cells than in the wild-type cells, and the expression of this protein in both cell lines was increased by treatment with 20 nM TGFβ1. (**B**) The formation of LC3 particles in wild-type and *ATG5* mutant cells. Both cell lines were transfected with *GFP-PDCD4* plasmid and stained with anti-LC3 antibody (white color). The green and blue colors indicate ectopic GFP-PDCD4 and the nucleus, respectively. Particles of LC3, an autophagosome marker, were seen even under normal culture conditions and were further increased under serum-starved conditions in wild-type Huh7 cells. However, these particles were not observed in *ATG5* mutant cells, even after autophagy induction by serum withdrawal.

**Figure 4 cells-09-00218-f004:**
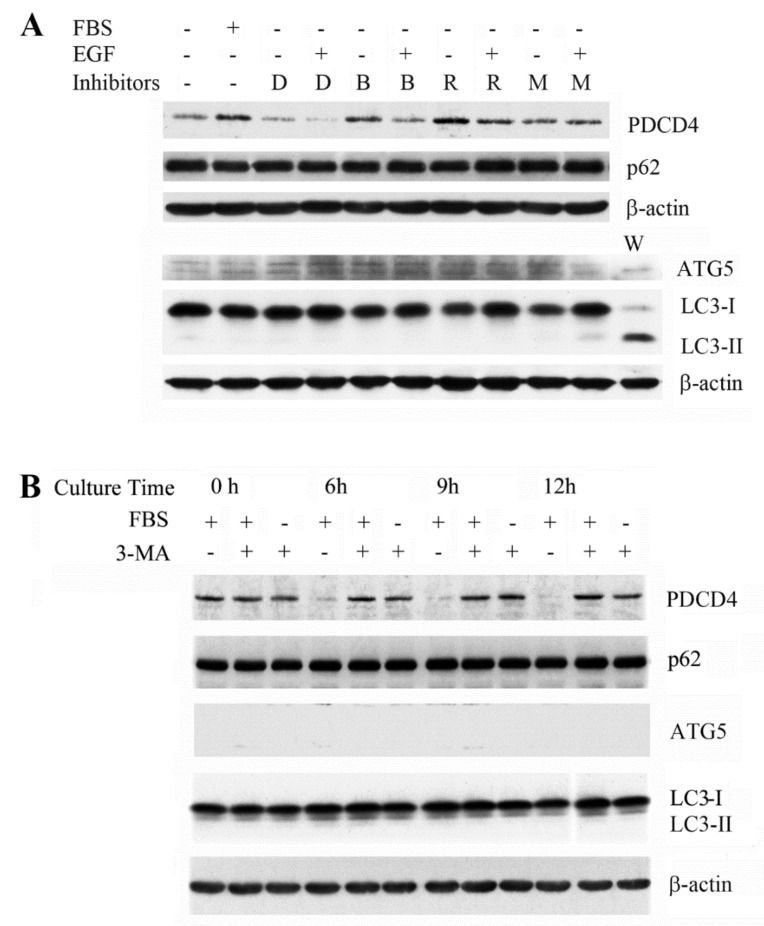
The PDCD4 levels were upregulated by the autophagy inhibitors bafilomycin A1 and 3-MA and by the proteasome inhibitor MG132 in *ATG5* mutant cells. (**A**) After reaching 90–95% confluency, the *ATG5* mutant Huh7 cells were treated with DMSO (D) as a control, 10 µM bafilomycin A1 (**B**), 0.1 nM rapamycin (R) and 20 µM MG132 in the presence or absence of EGF in normal or serum-free medium and incubated for 4 h. The cells were collected and Western blotting analyses were performed by using antibodies against the components mentioned in the figure. (**B**) *ATG5* mutant Huh7 cells at 90–95% confluency were treated with 5 mM 3-MA and the culture continued for 6, 9, and 12 h, after that cells were collected for Western blotting. Experiments were repeated at least three times and similar results were obtained. The representative results are demonstrated.

**Figure 5 cells-09-00218-f005:**
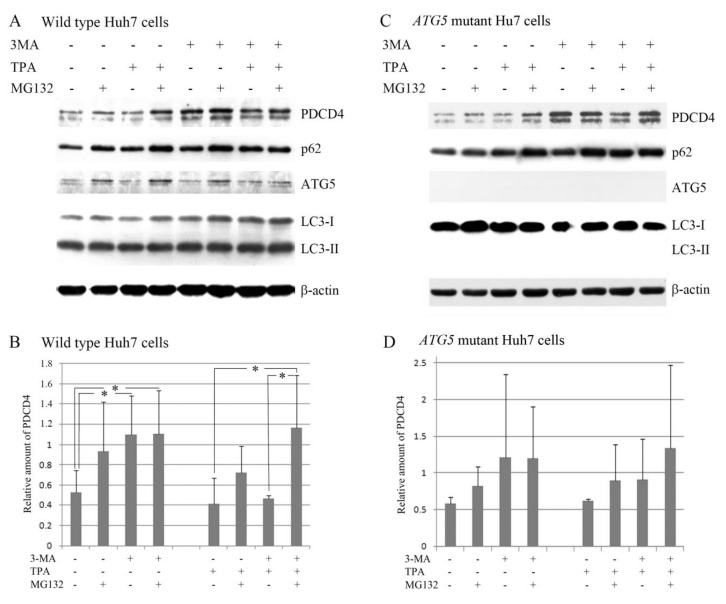
The PDCD4 levels were upregulated synergistically by 3-MA and MG132 in the wild (**A**,**B**) and *ATG5* mutant-16 (**C**,**D**) Huh7 cells. (A) After culturing for 4 days, Huh7 cells were treated with or without 5 mM 3-MA in the presence or absence of 50 µM TPA (12-*O*-tetradecanoylphorbol-13-acetate) and/or 20 µM MG132. TPA and MG132 were added to the culture at 1.5 h after the addition of 3-MA, and the cells were further cultured for another 4 h. Western blotting analyses were performed using antibodies against the components indicated in the figure. In this experiment, the mouse monoclonal anti-PDCD4 antibody (M-mAb) was used. This M-mAb detected two bands that might be PDCD4 protein, as both bands disappeared following PDCD4 knockdown. The lower band might be a degraded product of the upper band, as the lower band was unable to be detected by antibodies against the PDCD4 N terminal sequences. The experiments were repeated three times and the representative results of them is shown (B) A diagram of the amount of PDCD4 obtained by the evaluation of PDCD4 bands shown in (A). Data are shown as an average with SD of three independent experiments. The upper band was used to determine the PDCD4 levels that were normalized by β-actin. (C,D) Western blotting analyses (C) and a diagram of the amount of PDCD4 (D) obtained based on the PDCD4 band in (C) of *ATG5* mutant-16 Huh7 cells. Experiments were performed in the same way as described in (A) and (B). Data are shown as an average with SD of three independent experiments. Statistically significant differences (*p* < 0.05) are represented by asterisk symbol (*). In the case of *ATG5* mutant cells, although the inhibitors consistently upregulated PDCD4 levels the statistical significance was not obtained because of the large variations from culture to culture.

**Figure 6 cells-09-00218-f006:**
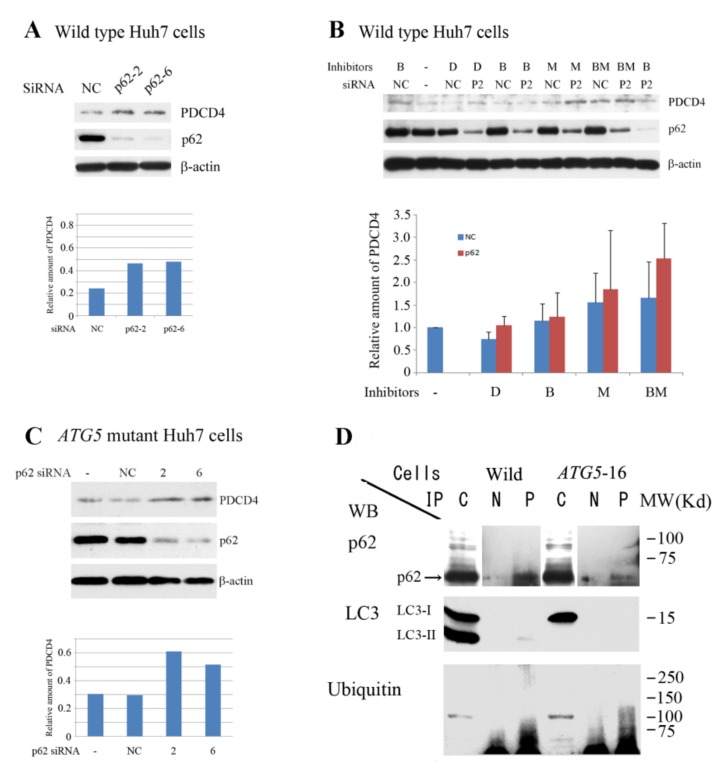
p62 was involved in PDCD4 degradation. (**A**) Knockdown of p62 upregulated the PDCD4 levels. Knockdown of p62/ SQSTM1 in Huh7 cells was performed using two different SQSTM1 siRNAs (p62-2 and p62-6). One dish was treated with negative control (NC) siRNA as a control. Upper panel, Western blotting analyses. Lower panel, diagram of the amount of PDCD4 obtained based on the PDCD4 bands in the upper panel. This experiment was performed twice, and similar results were obtained repeatedly. (**B**) After 24 h of p62 knockdown, the cells were treated with different inhibitors: DMSO (D), bafilomycin A1 (B), MG132 (M), and bafilomycin A1 and MG132 (BM), as mentioned in Figure 4. Upper panel, Western blotting analyses. The experiment was repeated three times, and a representative figure is shown. Lower panel, diagram of the amount of PDCD4 obtained from the Western blotting analyses. Although MG132 (M) as well as bafilomycin A1 and MG132 (BM) upregulated the PDCD4 levels, bafilomycin A1 (B) alone was unable to upregulate this protein in p62 knockdown cells. (**C**) p62 knockdown also upregulated the PDCD4 levels in *ATG5* mutant cells. The same protocol described in (A) was followed for the knockdown of p62 in *ATG5* mutant cells using p62 siRNAs 2 and 6. This experiment was performed twice, and similar results were obtained repeatedly. (**D**) p62 and LC3-II were co-immunoprecipitated with PDCD4. Extracts of wild and *ATG5* mutant Huh7 cells were immunoprecipitated with anti-PDCD4 antibody and the precipitates were analyzed by Western blotting using anti-p62, LC3 and ubiquitin antibodies. p62 and ubiquitin bands were observed in both cell types, but no LC3-II band was observed in the precipitate of mutant cells. Experiments were repeated more than three times and the representative figure is shown.

**Figure 7 cells-09-00218-f007:**
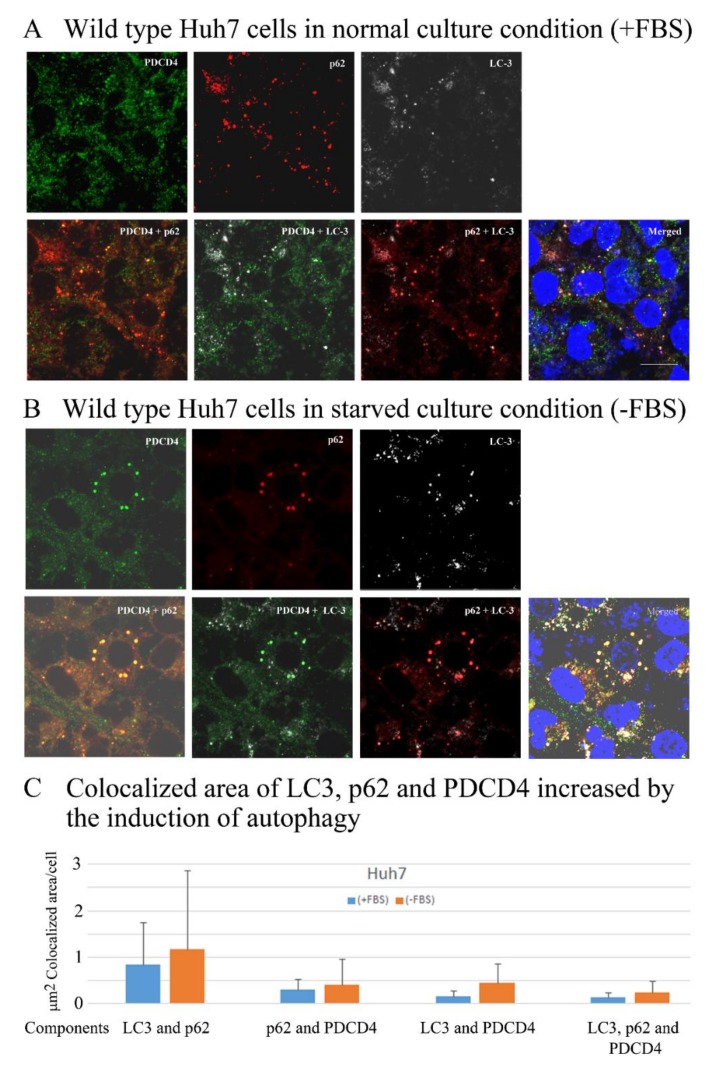
PDCD4, p62. and LC3 were colocalized, and the colocalization areas expanded under serum-deprived conditions in wild-type Huh7 cells. (**A**,**B**) A total of 1.5 × 10^5^ cells were seeded onto a cover glass in 35-mm dishes and cultured until 70–80% confluency. The old medium was replaced with Dulbecco’s modified Eagle’s medium (DMEM) or DMEM + 10% FBS. The dishes were washed twice with DMEM before adding the new medium. Culture was continued for 4 h, and the cells were fixed with 4% paraformaldehyde. Immunostaining of the cells were performed as described in methods. Images were captured using an LSM-880 confocal microscope. The colocalization of PDCD4, p62, and LC3 in the presence (A) and absence (B) of FBS is shown. (**C**) A diagram of the colocalization areas of PDCD4, p62, and LC3 in wild-type Huh7 cells. Bar = 300 µm.

**Figure 8 cells-09-00218-f008:**
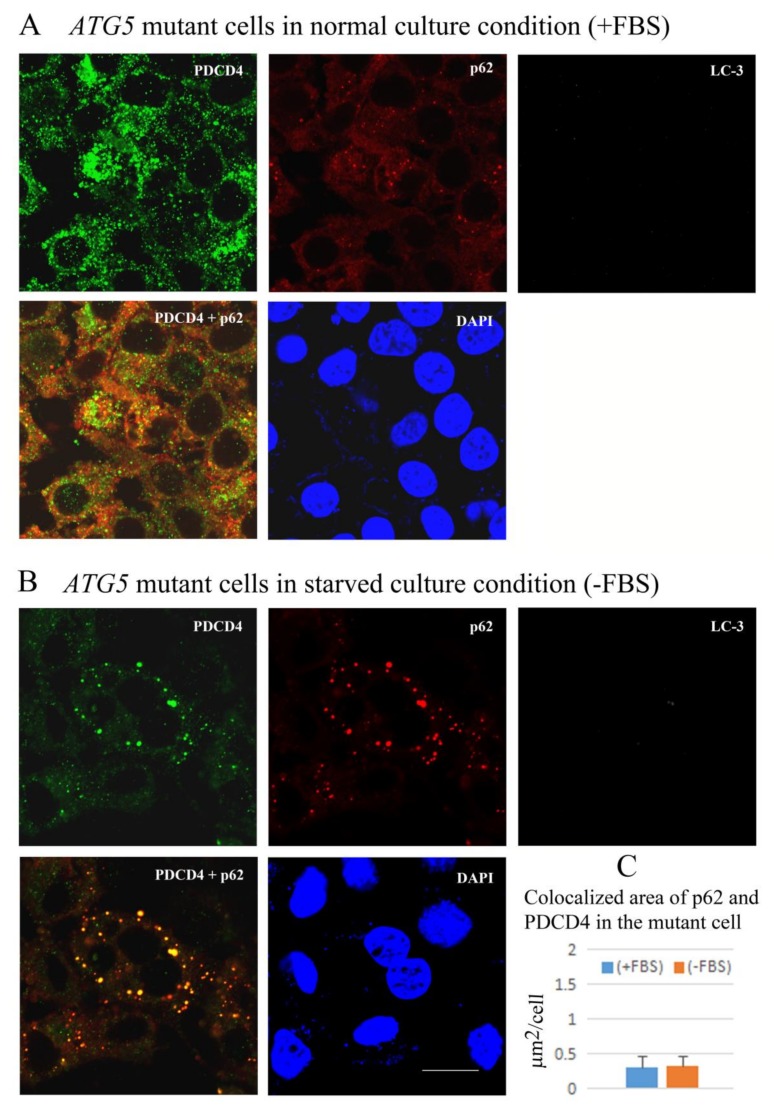
PDCD4 and p62 were colocalized, and the areas were slightly increased by serum starvation. The figures depicted the colocalization analyses of *ATG5* mutant-16 Huh7 cells under normal (**A**) and serum-starved culture conditions (**B**). The same protocol described in Figure 7 was used for this experiment. No LC3 particles were observed in this mutant cell line, even in the absence of FBS (B). (**C**) A diagram of the colocalization areas of PDCD4 and p62. Bar = 300 µm.

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
