# Peer review of "Degradation of the Tumor Suppressor PDCD4 Is Impaired by the Suppression of p62/SQSTM1 and Autophagy"

_cells, 2020, doi:10.3390/cells9010218_

Round 1
Reviewer 1 Report
In this manuscript the authors investigate the relationship between autophagy and the PDCD4 protein, which is accepted to be a tumor suppressor. They report that autophagy pathways, especially the P62/SQSTM pathway degrades the PDCD4 protein.
The manuscript depends largely on in vitro culture system utilizing cell lines that are related to autophagy as well as the western blot analyses of relative changes in protein expression levels.
I have some major questions and concerns.
In Figure 1,2,3, and 4 it is not clear how many times the experiments are performed, if the western blot is a representative of the other experiments, if there are any variations from sample to sample or experiment to experiment. There are no quantifications and no statistical analyses. There are some attempts for quantification as demonstrated in Fig 5 and 6, but the deviation is very large in each experimental group and there does not seem to be any statistical significance among groups. In Fig 6c, there is not even an indication for a standard error the mean (was that experiment performed once?).
Overall, I think reproducibility is lacking. Each western blot needs to be repeated at least 3 times and there needs to be some statistical analyses to show the quantitative differences. As is, we are forced to believe the intensity of a single band, but does not band represent the result of one sample, one experimental readout. In cases, where statistical analyses are performed the variation from sample to sample is so high that there does not seem to be a significant difference among groups.
There are immunocytochemistry results in cell lines, but I think it would be really nice to have expression level analyses in real tumor tissue, if this is related to cancer. The way the abstract and the introduction is written is confusing. One needs to read till the last sentence of the introduction to find out what the goal of the study is. I think it would be nice to state the problem in the field up front, define the question or the problem first, and then explain how the authors think to solve that problem or the missing information. Based on the data they show, there does not seem to be very significant difference, but they conclude to have an important impact and direct correlation. I think it is important to bring quantitation to analyses and to perform proper statistics to prove direct correlation and link. It would also be required to look at the expression levels of PDCD4 in real tumors. Is it downregulated or is it upregulated? Also in the same tumors how is the expression profile of P62, LC3-II and other markers they used in this study? Is this really correlation or causative?
Author Response
Answers to the Reviewer 1
cells-663520
Thank you for the reviewer’s kind and valuable comments. According to the reviewer’s suggestion, we have modified the manuscript. The modified portions were written in red.
>In Figure 1,2,3, and 4 it is not clear how many times the experiments are performed, if the western blot is a representative of the other experiments, if there are any variations from sample to sample or experiment to experiment. There are no quantifications and no
statistical analyses There are some attempts for quantification as demonstrated in Fig 5 and 6, but the deviation is very large in each experimental group and there does not seem to be any statistical significance among groups. In Fig 6c, there is not even an indication
for a standard error the mean (was that experiment performed once?).
Answer
Experiments were conducted independently at least three times except Figure 3A, Figure 6A and 6C, similar results were obtained from each experiment and the representative results were demonstrated. According to the reviewer’s comments, we have added explanation.
Figure 1. We have added the below in the figure legends.
“Experiments were repeated at least three times and similar results were obtained from each experiment. A representative result was shown in the figure.”
Figure 2. We have added the below in the figure legends.
“Experiments were repeated more than three times and similar results were obtained consistently in each case. The representative results were demonstrated.”
Figure 3A. The experiments were performed to isolate ATG5 deficient cells by CRIPR/Cas9 method. Western blot was repeated twice to confirm the diminished expression of ATG5 and the inhibited generation of LC3-II and almost same results were obtained. Consistent results of Western blot in ATG5 mutant cells were also shown in Figure 4.
Figure 4. We have added the below in the figure legends.
“Experiments were repeated at least times and similar results were obtained. The representative results were demonstrated.”
Figure 5. Statistical analysis was performed and significant differences were shown by asterisks symbol(Fig 5C). In Fig 5D from the results of ATG5 mutant cells, we could not see the significant statistical differences probably because of large variations while the consistent relative upregulation by inhibitors were observed. We added the below description in the figure legends.
“Data were shown as an average with 2SD of three independent experiments. Statistically significant differences (p<0.05) were represented by asterisk symbol (*). In the case of ATG5 mutant cells, although the inhibitors consistently upregulated PDCD4 levels but the statistical significance was not obtained because of the large variations from culture to culture.”
Figure 6. The experiments shown in Fig 6A and 6C were performed twice to select the most efficient siRNA to knockdown p62 and to see the involvement of p62 in the regulation of PDCD4 protein level. The description below was added in figure legends of Fig6A and 6C.
“This experiment was performed twice and similar results were obtained repeatedly.”
>There are immunocytochemistry results in cell lines, but I think it would be really nice to have expression level analyses in real tumor tissue, if this is related to cancer.
Answer
We agree to the reviewer’s opinion. As we mentioned in the manuscript, immunohistochemical study of PDCD4 expression in cancers were previously reported and the downregulation of PDCD4 was shown by several authors including us. Similarly, the increased expression of p62 and activation of autophagy in cancer tissue has been shown in several reports which was cited in the manuscript. In this manuscript, we first investigated the PDCD4 protein regulation by p62-mediated machinery to understand the detailed mechanisms. As the reviewer suggested, the further study using real tumor tissue is required to understand the mechanisms and relationships between PDCD4, p62 and LC3 in vivo.
>The way the abstract and the introduction is written is confusing. One needs to read till the last sentence of the introduction to find out what the goal of the study is. I think it would be nice to state the problem in the field up front, define the question or the problem first, and then explain how the authors think to solve that problem or the missing information.
Answer
We wrote the manuscript based on our previous studies on PDCD4 regulation mechanisms. According to the reviewer’s suggestion, we modified some parts of abstracts and introduction. The modified portion is shown in red character. We want the reviewer’s understanding about the stream of our previous work on PDCD4 protein regulation mechanisms that promoted us to investigate the PDCD4 degradation mechanisms by autophagy.
>Based on the data they show, there does not seem to be very significant difference,
but they conclude to have an important impact and direct correlation. I think it is important to bring quantitation to analyses and to perform proper statistics to prove direct correlation and link.
Answer
We agree to the reviewer’s comments. As we mentioned in reply to first question, experiments were conducted independently at least three times except Figure 3A, Figure 6A and 6C, and similar results showing the consistent changes of PDCD4 expression were obtained from each experiment.
>It would also be required to look at the expression levels of PDCD4 in real tumors. Is it downregulated or is it upregulated?
Answer
As we replied to previous question, PDCD4 expression is downregulated in many types of real tumors and the reports were cited and described in the manuscript.
>Also in the same tumors how is the expression profile of P62, LC3-II and other markers they used in this study? Is this really correlation or causative?
Answer
The upregulation of p62 and LC3-II were reported in previously and we cited and described those reports in the manuscript. However, the correlation or causative of these phenomena are still unclear and further study is necessary,.
Reviewer 2 Report
The manuscript by Manirujjaman et al. is very interesting and mostly well written. However, the data do not fully support the conclusion that degradation of PDCD4 "is controlled by p62/SQSTM-mediated autophagy". Actually, the degradation of PDCD4 is NOT blocked by the deletion of Atg5, meaning that conventional autophagy does not control PDCD4.
The "autophagy inhibitors" bafilomycin A1 and 3-MA do not specifically inhibit autophagy. They rather block processes that are important for autophagy AND other pathways. Therefore, it is not possible to prove ATG5-independent autophagy. Maybe it is ATG5-independent autophagy but other options are possible. The authors should rephrase the text accordingly.
To determine the mechanism of changes in the level of PDCD4, transcription should be investigated. Please perform quantitative RT-PCRs of PDCD4 to test whether the treatments change synthesis of PDCD4.
At several points of the manuscript, the authors write "expression of PDCD4" when they actually refer to the "level of PDCD4". Autophagy degrades the protein whereas gene EXPRESSION generates a protein. Please do not use the EXPRESSION or EXPRESSED to say "present".
Author Response

(The authors gave the same response as above.)

Round 2
Reviewer 2 Report
The manuscript has been improved but the conclusion and the title should be stated with more caution. Deletion of Atg5 does not block the degradation of PDCD4 and the active role of non-canonical pathway of autophagy is not proven. Please consider the title: Degradation of the tumor suppressor PDCD4 is impaired by the suppression of p62/SQSTM and autophagy.
The legends of the Supplementary Figures are missing.
Author Response
Answers to the Reviewer2
Thank you for your kind comments. According to the reviewer’s suggestion we have modified the manuscript. The modified portion in first revised version is written in red and the portion modified in the second-revised version is written in green.
>The manuscript has been improved but the conclusion and the title should be stated with more caution.Deletion of Atg5 does not block the degradation of PDCD4 and the active role of non-canonical pathway of autophagy is not proven. Please consider the title: Degradation of the tumor suppressor PDCD4 is impaired by the suppression of p62/SQSTM and autophagy.
Answer
According to the reviewer’s suggestion we have changed the title to “Degradation of the tumor suppressor PDCD4 is impaired by the suppression of p62/SQSTM and autophagy”.
We also modified the conclusion part as follows(Line 506 to 510 in the second revised manuscript):
“We demonstrated for the first time that the tumor suppressor PDCD4 is degraded by the p62-mediated selective macroautophagy system in Huh7 hepatoma cells. The autophagy system may contribute at least partly to suppress the levels of PDCD4 and result in the development and progression of tumor cells. Thus, the inhibition of this pathway might be a potential target in cancer therapy”.
>The legends of the Supplementary Figures are missing.
Answer
We have replaced supplementary figures with figure legends.
The modified portion in first revised version is written in red and the portion modified in the second-revised version is written in green.